# *Xanthoceras sorbifolia* Husk Extract Incorporation for the Improvement in Physical and Antioxidant Properties of Soy Protein Isolate Films

**DOI:** 10.3390/foods12152842

**Published:** 2023-07-27

**Authors:** Yingying Han, Wentao Yan, Yuping Hou, Dongmei Wang, Miao Yu

**Affiliations:** 1College of Chemistry & Pharmacy, Northwest A&F University, Yangling 712100, China; hyying@nwafu.edu.cn; 2College of Forestry, Northwest A&F University, Yangling 712100, China; yanwentao@nwafu.edu.cn (W.Y.); 1623205931@nwafu.edu.cn (Y.H.); dmwli@nwafu.edu.cn (D.W.); 3Shaanxi Key Laboratory of Economic Plant Resources Development and Utilization, Northwest A&F University, Yangling 712100, China

**Keywords:** soy protein isolate, plant extracts, antioxidants, films, food packaging, mechanical properties

## Abstract

With the increasing awareness of ecological and environmental protection, the research on eco-friendly materials has experienced a considerable increase. The objective of our study was to explore a novel soy protein isolate (SPI) film functionalized with antioxidants extracted from *Xanthoceras sorbifolia* husk (XSHE) as bio-based active packaging films. The films were evaluated in light of their structure, physical machinery, and antioxidant performance using advanced characterization techniques. The FTIR and microscopy results revealed the hydrogen-bond interaction between the SPI and XSHE and their good compatibility, which contributed to the improvement in various properties of the composite films, such as tensile strength (TS), UV blocking, and the water barrier property. As the XSHE content increased to 5%, the TS of the films dramatically increased up to 7.37 MPa with 47.7% and the water vapor permeability decreased to 1.13 × 10^−10^ g m m^−2^ s^−1^ Pa^−1^ with 22.1%. Meanwhile, the introduction of XSHE caused further improvement in the antioxidant capacity of films, and the release of active agents from films was faster and higher in 10% ethanol than it was in a 50% ethanol food simulant. Overall, SPI-based films functionalized with XSHE demonstrated promising potential applications in food packaging.

## 1. Introduction

Currently, petroleum-based synthetic plastics, including polystyrene, polypropylene, polycarbonate, and polyethylene, are widely used in food packaging industries [1]. Nevertheless, with the increasing awareness of environmental protection and food safety, people are seeking substitutes to alleviate environmental pollution, preserve fossil resources, and guarantee food quality [2,3]. Bio-packaging is recognized as an alternative packaging, which aims to reduce and simplify food packaging materials and create innovative applications [4]. Many researchers put much effort into developing biomass-based edible films with excellent mechanical strength and water and gas barrier performance [5,6]. Biopolymers from natural resources (such as polysaccharides, proteins, lipids, and resins) are usually employed as the matrices of food packaging films [2,7]. Among the different varieties of biopolymers, protein has drawn great attention due to its merits, such as low-cost, abundance, edibility, and biodegradability [8]. As a common by-product of soybean oil production industries [9], the film-forming property of the soy protein isolate (SPI) is widely investigated. The main components of the SPI are albumins and globulins, containing eighteen different amino acids [10,11]. Polar amino acids (e.g., arginine, lysine, cysteine, and histidine) can be used to establish the cross-linked protein network to improve the thermal and physical properties. Typically, SPI-derived films have good light transmittance, excellent gas barrier performance, and moderate mechanical properties and can be widely used in food packaging and coating areas, offering an attractive alternative to replace petroleum-based materials [3,12].

Bio-based films, such as SPI-based films, can serve as the carriers of bioactive substances (antioxidants and antibacterial agents) to produce active packaging (AP) films [13,14,15,16]. AP films prevent food deterioration and prolong the shelf life, which are sensitive to oxidation and microbiological effects [17,18]. The activity of the amino acid side chains in SPI molecules can offer multiple sites by absorbing or releasing active compounds for the fabrication of SPI-based packaging films [19,20]. Due to the health demands of customers, studies on active packaging tend to use natural active compounds or extracts rich in antioxidant and antibacterial components, instead of synthetic ingredients [21,22]. Recently, various kinds of natural ingredients have been utilized to improve the antioxidant and antibacterial properties of SPI films, including cortex phellodendron extract [23], tannins [24,25], essential oils [17,26], and mango kernel extract [27,28].

*Xanthoceras sorbifolia*, also known as yellow horn, is an important oil crop native to northwest China, belonging to the family Sapindaceae [29,30]. The seed kernel of *Xanthoceras sorbifolia* has an oil content of more than 50%, and it is rich in unsaturated fatty acids, such as linoleic and oleic acids [31,32], which are favorable for medicinal, nutritional, and industrial applications. Currently, the interest in *Xanthoceras sorbifolia* mainly focuses on oil extraction from the seed kernel, a process that generates quantities of residues, including husks and seed meals. In recent years, the biological activities of *Xanthoceras sorbifolia* husks have drawn great attention. Chemical studies indicated that the husks contain multiple compounds, such as polyphenol, triterpenoid, and sterol [33]. A pharmacological study revealed that the husk components of *Xanthoceras sorbifolia* showed many bioactivities, such as antitumor [34], tyrosinase inhibitory [35], memory improvement [36], anti-inflammatory, and neuroprotective effects [37]. Ling et al. [38] analyzed two new triterpenoid glycosides from the ethanol extracts of the husks using an HPLC-ESI-MS analytical method. Yang et al. [33] successfully isolated thirty-seven phenolic compounds from *Xanthoceras sorbifolia* husks and suggested that husk extracts had useful antitumor and radical scavenging activities. Thus, the incorporation of active extracts from *Xanthoceras sorbifolia* husks as antioxidant additives in food packaging is a novel concept to improve the comprehensive utilization of the residue. To our knowledge, there was no combination of the SPI and *Xanthoceras sorbifolia* husk extract (XSHE) for the fabrication of active films.

The purpose of the present work was to prepare SPI-based active packaging films with XSHE loading via the casting method. The effects of incorporating XSHE on the structure, mechanical performance, light transmittance, water vapor permeability, and antioxidant activity of the films were systematically investigated. In addition, the release behavior of films in two different food simulants was also studied.

## 2. Materials and Methods

### 2.1. Materials

The SPI was bought from Shansong Biological Products Co., Ltd. (Linyi, China). *Xanthoceras sorbifolia* husks were collected in August 2021 in Xinzhou city, Shanxi Province. Sodium hydroxide (NaOH) and potassium carbonate (K_2_CO_3_) were provided by Guanghua Sci-Tech Co., Ltd. (Guangzhou, China). Ethanol and glycerol were ordered from Kelong Chemical Co., Ltd. (Chengdu, China). The ABTS reagent (2,2′-azino-bis [3-ethylbenzothiazoline-6-sulfonic acid] diammonium salt) was provided by Sigma‒Aldrich Inc. (Saint Louis, MO, USA). The DPPH reagent (2,2-diphenyl-1-picrylhydrazyl) was purchased from Alfa Aesar (Ward Hill, MA, USA). The Folin–Ciocalteu reagent was provided by Solarbio Inc. (Beijing, China).

### 2.2. Extraction of XSHE

The extraction method was according to the study of Yang et al. (2020) [39] with slight modification. The crushed husks (0.5 kg) were extracted 2 times with 95% aqueous ethanol (500 mL) until reflux and maintained at 80 °C for 3 h and 2 h, respectively. The extracting solution was filtered and concentrated in a vacuum at 50 °C for ~2 h to yield a thick liquid extract. The *Xanthoceras sorbifolia* husk extract (XSHE) with a 0.86 g g^−1^ concentration was stored at 4 °C.

### 2.3. Formation of SPI Films

The SPI (6.0 g) and glycerol (3.0 g) were dissolved in deionized water (100 mL) with continuous stirring. The uniform solution was adjusted with NaOH (2.0 M) to pH ≈ 9 and stirred at 85 °C for 20 min [3]. Thereafter, XSHE (0, 1, 3, 5, and 7% *w*/*w* of SPI content) dispersed in 5 mL of 50% ethanol was incorporated into the cooled solution with stirring for 10 min. The mixed solution was poured into a polymethyl methacrylate (PMMA) plate (26 cm × 26 cm × 4 cm) and then dried at 55 °C overnight in a vacuum oven. Finally, the films were peeled from the plates and kept at 43 ± 2% relative humidity (RH) for further analyses. According to the additive amount of XSHE, the obtained films were designated as XSHE-0, XSHE-1, XSHE-3, XSHE-5, and XSHE-7.

### 2.4. Characterization of Films

#### 2.4.1. Scanning Electron Microscopy (SEM)

The morphologies of the samples sputter-coated with gold were characterized using a scanning electron microscope (S-4800, Hitachi Co., Tokyo, Japan, 10 kV) [3].

#### 2.4.2. Fourier Transform Infrared (FT-IR) Spectroscopy

The FT-IR spectra of the XSHE and films were characterized using a FT-IR spectrometer (Nicolet 6700, Thermo Fisher Scientific, Waltham, MA, USA). The spectra were collected from 32 scans in the range of 4000–650 cm^−1^ at a resolution of 4 cm^−1^ [16].

#### 2.4.3. Colorimetric Analysis

The color of the films was measured using a portable colorimeter (3NH TS 20, Shenzhen, China). The total color difference (Δ*E*) was measured as follows [5]:$$\Delta E=\sqrt{{\left(\Delta L\right)}^{2}+{\left(\Delta a\right)}^{2}+{\left(\Delta b\right)}^{2}}$$
where Δ*L* = *L* − *L**; Δ*a* = *a* − *a**; Δ*b* = *b* − *b**; and *L*, *a*, and *b* are the color parameters of the film samples. *L**, *a**, and *b** are the color parameters of the standard plate.

#### 2.4.4. Light Transmittance

An ultraviolet-visible (UV-vis) spectrophotometer (Shimadzu-1780, Kyoto, Japan) was used to characterize the light transmittance of films at wavelengths between 200 and 800 nm [18].

#### 2.4.5. Mechanical Performance

The mechanical performance of the film samples was analyzed using an electronic tensile machine (Labthink BLD-200N, Ji’nan, China). Tensile strength (TS) and elongation at break (EB) are two indicators to evaluate the mechanical properties. First, the films were tailored to rectangular strips (15 mm × 80 mm), and the thickness was recorded at five arbitrary positions using a handheld electronic micrometer (Model MDC-25, Huzhou, China). The initial grip separation was 50 mm, and the crosshead speed was set at 300 mm min^−1^ [22]. At least five specimens of each sample were measured in parallel.

#### 2.4.6. Moisture Content (MC) and Water Vapor Permeability (WVP)

The moisture content (MC) of the samples was tested using a gravimetric method after drying at 105 °C. The determination of WVP was conducted via a modified gravimetric method according to previous work [40]. In brief, the films were sealed into weighing bottles containing anhydrous CaCl_2_ (0% RH) using hot melt adhesive. Thereafter, the bottles were kept in a desiccator at 25 °C containing a saturated sodium chloride solution (75% RH). Then, the bottles were weighed at regular intervals for 48 h.

#### 2.4.7. Release Test

The total phenolic content (TPC) release of films was performed using the Folin–Ciocalteu method [41]. First, pre-weighed films (100 mg) were immersed in 5 mL of 10% (*v*/*v*) (water-based food simulant) and 50% (*v*/*v*) ethanol (fatty food simulant) at 25 °C in a shaker (100 rpm) to obtain the film solution. Multiple samples under this condition were tested simultaneously. Then, the film solution was drawn out from the flasks in turn at different times. The mixture of the film solution (1 mL), the sodium carbonate solution (5 mL, 0.7 mol L^−1^), and the Folin–Ciocalteu reagent (4 mL, 0.1 mol L^−1^) was dark-incubated for 120 min at 25 °C, and then, the absorbance was measured at 765 nm. The TPC result was expressed as gallic acid equivalents per gram (mg g^−1^) of the dried film according to the following formula:$$\mathrm{TPC}=\frac{C\times V}{m}$$
where *C* (mg L^−1^) represents the concentration of gallic acid calculated from the calibration curve, *V* (L) refers to the testing film solution volume, and *m* (g) is the mass of the film.

#### 2.4.8. Antioxidant Activities

The antioxidant activities were measured using the DPPH assay and the ABTS assay, as described by Ref. [21], with slight modification. In the DPPH assay, 0.4 mL of the film solution was reacted with 3.6 mL of the DPPH–ethanol reagent (25 mg L^−1^) for 0.5 h at 25 °C in a dark environment, and then, the absorbance was determined at 517 nm. The DPPH scavenging activity was measured using the following equation:$$\mathrm{DPPH}\;\mathrm{radical}\;\mathrm{scavenging}\;\left(\%\right)=\frac{A_{c}-A_{s}}{A_{c}}\times 100$$
where *A_c_* and *A_s_* denote the absorbance of the control and the testing solution at 517 nm, respectively.

In the ABTS method, the working solution of ABTS was obtained by mixing the ABTS solution (7 mM) with potassium persulfate (2.45 mM) for 16 h without light, then an absorbance of 0.70 ± 0.02 was achieved at 751 nm by diluting with ethanol. Thereafter, a 0.3 mL aliquot of the film solution and 3.7 mL of the diluted ABTS solution were reacted for 10 min at 25 °C in the dark. A UV–vis spectrophotometer was used to measure the absorbance of the incubated solution at 751 nm. The ABTS radical scavenging activity was computed by using the following equation:$$\mathrm{ABTS}\;\mathrm{radical}\;\mathrm{scavenging}\;\left(\%\right)=\frac{A_{c}-A_{s}}{A_{c}}\times 100$$
where *A_c_* and *A_s_* represent the absorbance of the control and the testing solution at 751 nm, respectively.

#### 2.4.9. Statistical Analysis

All experiments were conducted at least in triplicate, and the data were expressed as the mean ± standard deviation. An analysis of variance (ANOVA) and Duncan’s multiple range test were performed to assess the significant differences (*p* < 0.05) between the films using SPSS Statistics 19 software (Chicago, IL, USA) [23].

## 3. Results and Discussion

### 3.1. Morphology of the Films

The morphologies of the films were studied via surface and cross-sectional observation, as shown in Figure 1. The control SPI film (XSHE-0) had a compact, smooth, and clean surface. The films with XSHE (Figure 1b–d) exhibited a very similar morphology to the control film, while the surface of the cross-sections became somewhat rougher, probably evidencing the hydrogen bond interaction between the SPI and XSHE. There were small pores and grooves on the films when the XSHE content increased to 7%, which was attributed to the irregular distribution of XSHE in the SPI matrix and the reduced compatibility between them. Similar observations were obtained in the whey protein isolate-based films incorporating curcumin [42].

### 3.2. FT-IR Spectra

The ATR-FTIR spectra of the XSHE and SPI films are presented in Figure 2. The band of XSHE at approximately 3290 cm^−1^ was associated with -OH stretching. The peak at 2928 cm^−1^ was assigned to the stretching vibrations of -CH_2_ groups. The peak at 1710 cm^−1^ arose from the carbonyl group. The peak at 1043 cm^−1^ was due to the characteristic C-O stretching vibration. The bands at 1605–1435 cm^−1^, especially two peaks at 1605 and 1508 cm^−1^, were ascribed to aromatic C=C stretching [43]. Many studies have investigated the main active components (including triterpenoids and polyphenols) of *Xanthoceras sorbifolia* husks (XSH) responsible for antitumor and radical scavenging activities. Wang et al. [44] established structures of ten barrigenol triterpenoids isolated from XSH and presented that the IR spectrum of these compounds showed the absorption bands of carbonyl (~1720 cm^−1^) and hydroxyl (~3400 cm^−1^) groups. Yang et al. [33] extracted and isolated 37 polyphenols from XSH and determined their structure–activity relationships, and the results showed that polyphenols usually have absorption peaks at around 3360 cm^−1^ (hydroxyl) and 2920 cm^−1^ (alkyl). The two peaks at ~1600 and 1500 cm^−1^ were of great significance for the determination of the aromatic nuclei structure. Hergert et al. [45] suggested that the 1600 cm^−1^ band was more intense than the 1500 cm^−1^ in phenolic extractives such as flavanones due to the presence of a phloroglucinol and catechol nucleus. Thus, the FTIR spectra of XSHE demonstrated that the main components were triterpenoids and polyphenols. Moreover, the lower wave number of hydroxyl groups of XSHE compared with those found in the literature may be due to the stronger hydrogen bonding between the various components of the extract while the components in their work were pure.

In the control SPI film, the absorption bands at 1628 cm^−1^, 1535 cm^−1^, and 1239 cm^−1^ were related to C=O stretching (amide I), N-H bending (amide II), and C-N stretching (amide III) vibrations, respectively [22,46]. The broad absorption band at 3268 cm^−1^ corresponds to the stretching vibration of the O-H and N-H groups [24]. The addition of XSHE resulted in a slight alteration in the intensity of the peaks in the spectrum of films, including decreased strength in the amide I and amide III bands of the SPI composite films, as well as in the peaks at 1040 cm^−1^ and 1390 cm^−1^, which confirmed the interaction of the hydroxyl groups of glycerol and XSHE with the main peptide chains of the SPI in the composite films through a hydrogen bond [47].

### 3.3. Film Color

Table 1 listed the color indices of the films with and without XSHE at various levels in terms of *L*, *a*, *b*, and Δ*E* values. The addition of XSHE reduced the L values as compared to the neat film, illustrating the decrease in lightness. The values of red–green (*a*) and blue–yellow (*b*) increased with the increasing XSHE concentration, implying that the color of the films gradually became redder and yellower, which were associated with the intrinsic color of XSHE. Among these color indices, a notable increase in the *b* values (from 9.84 to 22.65) was observed, indicating that the color of the films became significantly yellowish. Moreover, SPI films with XSHE showed a higher total color difference (Δ*E*) than the neat SPI film, implying that the films became more colored. Overall, the results suggested that the variance in the color of the SPI–XSHE films was closely connected with the concentration of XSHE. Other researchers reported similar results with SPI films incorporated with pine needle extract [48] and diatomite/thymol complexes [16].

### 3.4. Light Transmittance of SPI–XSHE Films

UV‒vis blocking is a crucial property in the applications of food packaging. Light exposure arouses safety and quality problems, including oxidation, degradation, nutrient destruction, and off-flavoring, especially for photosensitive products [22,49,50]. Thus, food packaging with excellent light barrier properties is often preferred. Figure 3 presented the effects of XSHE incorporation on the light transmittance of SPI films. Compared with a pure SPI film, the increase in XSHE content from 1 to 7% led to a significant decrease in the UV light transmittance (200–400 nm) of the SPI/XSHE composite films. The increase in the interaction between the SPI and XSHE and the light brown of XSHE enhanced the barrier ability of SPI films to ultraviolet light. Moreover, the high transmittance of all films for visible light indicated that the opacity of the films was not dramatically altered with the incorporation of XSHE. To summarize, the SPI films incorporating XSHE have a contribution to limiting UV-induced food spoilage due to its excellent UV-blocking properties.

### 3.5. Mechanical Properties

The mechanical properties in terms of TS and EB of different composite films were shown in Table 2. The XSHE-0 film exhibited the TS and EB of 4.99 MPa and 169.33%, respectively. For the composite films with XSHE, the TS presented a significant trend from rising to falling, while the changing trend of EB was the opposite. The TS of the XSHE-5 film had a maximum value of 7.37 MPa, and the EB reached a minimum value of 126.67%, which was ascribed to the stable reticulate structure establishment via cross-linking with the addition of XSHE. The cross-linking was produced by the interaction between the amino groups of protein and the phenolic groups of XSHE (such as flavonoids) via hydrogen bonds, decreasing the flexibility of the molecular chain and thus improving the rigidity of the film [28,43]. In addition, the polyphenols of XSHE have a stable cyclic structure, which also decreased the mobility of the SPI molecule [3]. It was necessary to increase the external pressure to break down the cross-linked structure when the composite film was stressed, thus resulting in an increased TS and a decreased EB. When the XSHE concentration was over 5%, the excess XSHE to the SPI led to a heterogeneous and discontinuous structure, resulting in a low TS value, which was ascribed to the excessive hydroxyl groups interacting with similar hydroxyl groups and reducing the attractive force [51]. The schematic formation mechanism of SPI-based active films with varying XSHE loading is proposed in Figure 1. The strength of the XSHE-5 film (TS: 7.37 MPa; EB%: 126.67%) was better than those of other reported SPI-based films, such as SPI/mango kernel extract films (TS: 3.06 MPa; EB%: 49.92%) [28], SPI/blueberry extract films (TS: 3.80 MPa; EB%: 37.30%) [52], and SPI/chestnut bur extract films (TS: 2.1 MPa; EB%: 201%) [24].

### 3.6. MC and WVP of SPI–XSHE Films

The MC of the SPI-based films was summarized in Table 3. The XSHE-0 film had the highest MC, and the MC of SPI films decreased with increasing XSHE content, attributed to the hydrophobic nature of XSHE. Food is more susceptible to deterioration when exposed to a high humidity circumstance, and the WVP of films is the key property for food packaging applications. Therefore, a film with an excellent barrier toward WVP is crucial. Based on Table 3, the WVP values decreased from 1.45 × 10^−10^ g m^−1^ s^−1^ Pa^−1^ to 1.13 × 10^−10^ g m^−1^ s^−1^ Pa^−1^ as the concentration of XSHE increased from 0% to 5%, which may be linked to the formation of hydrogen interactions in the film matrix. The hydrophobic nature of XSHE limited the availability of hydrogen groups to form hydrophilic bonds, thereby improving the water barrier property [53]. Notably, the WVP of films increased to 1.20 × 10^−10^ g m^−1^ s^−1^ Pa^−1^ with 7% XSHE compared to 5% XSHE, indicating that the permeability of the films depends on not only the chemical microstructure but also the morphology [20]. Thus, the excess XSHE (7%) in the film matrix destroyed the dense structure, which resulted in an increase in WVP. The result was in accordance with the SEM analysis. Moreover, the optimal WVP of the SPI/XSHE films (1.13 × 10^−10^ g m^−1^ s^−1^ Pa^−1^) was better than that of the SPI/mango kernel extract films (7.93 × 10^−8^ g m^−1^ s^−1^ Pa^−1^) [28] and the SPI/oregano extract films (2.67 × 10^−9^ g m^−1^ s^−1^ Pa^−1^) [54].

### 3.7. Food Simulant Release Behavior

Polyphenol compound release is considered to be a crucial characteristic during food preservation, and the release behavior of XSHE was investigated in two food simulants containing 10% and 50% ethanol. As presented in Figure 4, a similar release tendency of all samples was observed in both simulants with a quick release in the first stage, followed by a continuous slow releasing and reaching of the equilibrium state in the end. The TPC of the control SPI film at equilibrium was a 2.1 and a 1.7 mg gallic acid g^−1^ sample in 10% ethanol and 50% ethanol, respectively. It was due to the presence of amino acids such as tyrosine, tryptophan, cysteine, and histidine of the SPI that modified the amino acid structures via heat treatment and the small peptides in the SPI hydrolysates [55]. The XSHE loading remarkably increased the release of TPC from the films, giving a range of a 2.4–2.8 mg gallic acid g^−1^ sample and a 1.8–2.4 mg gallic acid g^−1^ sample in 10% ethanol and 50% ethanol, respectively. Buonocore et al. [56] reported that the release of active substances from polymer networks occurred in multiple stages. First, solvent molecules penetrated the polymer matrix from the simulated solution, giving rise to the swelling of network. The structural changes in the films facilitated the release of the active compounds from the polymer matrix to the simulated solution until equilibrium was reached. Therefore, the release of XSHE was affected by several factors, such as the solvent environment, polymer solubility, and intermolecular interactions between XSHE and the SPI.

As shown in Figure 4, the release of TPC in 10% ethanol was faster than that in 50% ethanol. For the XSHE-5 film, the equilibration time in 10% ethanol was 360 min and significantly faster than that in 50% ethanol, which was attributed to the hydrophilic characteristic of the SPI. The films swelled by absorbing water after exposure to the ethanol solution. In the 10% ethanol system, the polar water molecules easily penetrated the protein matrix and led to a looser structure of the internal network, which governed polyphenol release. In contrast, the equilibration time in 50% ethanol was 900 min, which was significantly slower than that in 10% ethanol. This result is consistent with the influence of solvent molecule polarity reported by Sánchez-González et al. [57]. In summary, the TPC release behaviors in the two food stimulants were illustrated by the polarity of solvent molecules.

### 3.8. Antioxidant Activities

The antioxidant activities of the films were determined using two types of antioxidant analyses (DPPH and ABTS radical scavenging assays). The antioxidant activities of the control film were attributed to the amino acids in the SPI. Similar results were reported by Adilah et al. [27]. The DPPH and ABTS radical scavenging activities (Figure 5) of the films incorporated with XSHE increased with an increasing amount of XSHE due to the major components of XSHE, including polyphenols (such as flavonoids, phenolic acids, and coumarins), triterpenoids, and sterols [33], which are high-quality sources of antioxidants. The obtained results are in accordance with the increasing scavenging activities of the SPI-based films incorporating mango kernel extract [28]. Moreover, blend film analysis showed higher antioxidant capacities as measured by the ABTS assay relative to the DPPH assay, which was due to the different application scopes of the ABTS (both hydrophilic and lipophilic antioxidant systems) and DPPH assays (hydrophobic systems) based on their characteristics [58]. XSHE not only enhanced the film strength but also improved the TPC and antioxidant activities. Thus, XSHE acts as both a reinforcement agent and antioxidant to enhance its functional properties as an active packaging material.

## 4. Conclusions

The novel SPI-based active films functionalized with XSHE were achieved to develop an effective antioxidative packaging system. The interaction between XSHE and the SPI matrix via hydrogen bonding was proven via FTIR, and their compatibility was observed using SEM. The tensile strength and barrier (water vapor and UV-light) properties of the composite films were improved due to XSHE loading. However, the excessive addition of XSHE (7% *w*/*w*) caused a heterostructure with a discontinuous area of the films, thus resulting in a higher WVP and a lower TS. The optimum film properties were demonstrated at a XSHE content of 5%. Furthermore, the TPC released in two different food simulants and antioxidant activities of the composite films were greatly enhanced as the concentration of XSHE increased. In summary, this work provides an innovative concept for the fabrication of SPI films, which exhibit potential promise in active food packaging. In future work, blending with reinforcing components and improved processing approaches such as chemical cross-linking will be conducted to further enhance the barrier performance of SPI active films that allow applications in water-containing food packaging.

## Data Availability

The data are contained within the article.

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
