# Peer review of "Xanthoceras sorbifolia Husk Extract Incorporation for the Improvement in Physical and Antioxidant Properties of Soy Protein Isolate Films"

_foods, 2023, doi:10.3390/foods12152842_

Round 1
Reviewer 1 Report (New Reviewer)
The manuscript entitled: "Development of bio-based films from soy protein isolate incorporating Xanthoceras sorbifolia husk extract: physicochemical, antioxidant and food simulant release properties" is in the field of active packaging well designed and appropriate for the journal. Some comments as follows recommend improving the quality of the manuscript:
1- Title: The title is too long and recommend making it short and informative.
2- Abstract: Is OK, I think it was revised.
3- Keywords: Choose keywords other than the main words in the title.
4- Introduction: It is long and needs to update the literature mostly for 2020+. You can find a few recent articles from MDPI to update the background for example this is interesting: https://www.mdpi.com/2073-4360/15/13/2889
5- Materials and Methods: It was already revised. Make sure all methods have a proper reference(s) from either a published article or a standard method.
6- Results and discussions: Well developed.
7- Conclusion: Too long, justify your hypothesis and future research recommendation (if any).
Author Response
Please see the attachment.

Reviewer 2 Report (New Reviewer)
Review
Manuscript ID: foods-2517541
Journal: foods
Development of bio-based films from soy protein isolate incorporating Xanthoceras sorbifolia husk extract: physicochemical, antioxidant and food simulant release properties
The author try to study explore a novel soy protein isolate (SPI) films functionalized with antioxidants extracted from Xanthoceras sorbifolia husk (XSHE) as bio-based active packaging films. The films were evaluated in the light of their structure, physical machinery, and antioxidant performance through advanced characterization techniques. FTIR and the microscopy results revealed the hydrogen-bond interaction between the SPI and XSHE and good compatibility of them, which contributed to the improvement of various properties of the composite films, such as tensile strength (TS), UV blocking, and water barrier property. As the XSHE content increased to 5%, the TS of the films dramatically increased up to 7.37 MPa with 47.7% and the water vapor permeability decreased to 1.13 × 10-10 g m m-2 s-1 Pa-1 with.1%. Meanwhile, the introduction of XSHE caused further improvement in the antioxidant capacity of films, and the release of active agents from films was faster and higher in 10% ethanol than its in 50% ethanol food simulant.
But there are some comments as the following:
1- The author do not show problem at abstract
2- The author must add more references to introduction part
3- The author must add more references to all parts of manuscript
4- The author must add the novelty of this work
5- The authors must modify keywords at least six words
6- Figure 1 ,the authors must add magnification power
7- FTIR spectra fig 2, the authors must rewrite units at fig where transmittance % is wrong because there are more than one peaks
8- The authors must add weathering accelerated test to work
Recommendation: major revision
need grammer revision
Author Response
Please see the attachment.

Reviewer 3 Report (New Reviewer)
In this study, the researchers investigated the potential of utilizing soy protein isolate (SPI) films incorporated with antioxidants derived from Xanthoceras sorbifolia husk (XSHE) as bio-based active packaging films. The films were subjected to comprehensive analysis, focusing on their structural properties, physical characteristics, and antioxidant performance, employing advanced characterization techniques.
The topic of the manuscript is both original and relevant to the field as it addresses a specific gap. Bio-based films, specifically SPI-based films, have the potential to be utilized as carriers for bioactive substances, such as antioxidants and antibacterial agents, in the production of active packaging (AP) films. These AP films play a crucial role in preventing food deterioration and extending shelf-life, particularly in terms of combating oxidation and microbiological effects. This study emphasizes the significance of alternative, sustainable, and environmentally friendly packaging materials, catering to health-conscious consumers. Moreover, in the realm of active packaging, there is a growing tendency to utilize natural active compounds or extracts that are abundant in antioxidant and antibacterial components, rather than relying on synthetic ingredients.
This study contributes to the subject area of "Food Packaging Preservation" by introducing a novel approach to developing bio-based active packaging films. The researchers explore the utilization of soy protein isolate (SPI) films that are functionalized with antioxidants extracted from Xanthoceras sorbifolia husk (XSHE). This approach offers several advancements compared to existing published material.
· Firstly, the study demonstrates the successful incorporation of XSHE antioxidants into SPI films, resulting in improved properties such as tensile strength, UV blocking, and water barrier capability.
· Secondly, the research provides insights into the interaction between SPI and XSHE, revealing a hydrogen-bond interaction and good compatibility between the materials.
· Additionally, the study investigates the antioxidant capacity of the films and examines the release of active agents in different ethanol concentrations.
Regarding the methodology, there are a few specific improvements and further controls that the authors could consider:
- Stability and durability assessment: To enhance the practicality of the SPI-XSHE films for food packaging, it would be valuable to investigate their stability and durability under different storage conditions, such as varying temperatures, humidity levels, and exposure to light. This would provide insights into the films' performance over time.
- Comparative studies: To better understand the advantages and limitations of the SPI-XSHE films, it would be beneficial to compare their properties and performance with existing packaging materials or other bio-based active films.
- Migration studies: Since the films are intended for food packaging applications, it would be important to assess the migration of XSHE antioxidants or any other components from the films into food simulants.
- Shelf-life extension evaluation: To demonstrate the practical utility of the SPI-XSHE films, conducting shelf-life extension studies using real food products would be valuable. This would involve assessing the films' ability to preserve the quality and extend the shelf life of various perishable food items.
The conclusions presented in the manuscript are consistent with the evidence and arguments provided throughout the study. The researchers successfully achieved SPI-based active films functionalized with XSHE, demonstrating the interaction between XSHE and the SPI matrix through hydrogen bonding, as confirmed by FTIR analysis. Compatibility between XSHE and SPI was also observed through SEM imaging.
The study showed that the incorporation of XSHE improved various properties of the composite films, including tensile strength, water vapor barrier, UV-light blocking, and antioxidant activities. However, the excessive addition of XSHE resulted in a heterostructure with discontinuous areas in the films, leading to higher water vapor permeability and lower tensile strength. The optimum film properties were achieved at a XSHE content of 5%.
The provided references for the study appear to be relevant and appropriate for the research topic. The references cover a wide range of aspects related to bio-based films, food packaging, and active packaging, which align with the subject area of the study. These references include studies on the properties, characterization, antimicrobial activity, antioxidant activity, and application of bioactive edible films based on various materials such as soy protein isolate, chitosan, cellulose nanocrystals, and green tea extract.
Please improve the resolution of the figures, particularly Figure 1, Figure 3, Scheme 1, Figure 4 and legend of figure 5. The authors should consider enhancing the image quality and ensuring that the figures are clear, legible, and visually appealing. Additionally, they should ensure that the labels, captions, and legends associated with these figures are accurately and clearly represented.
Author Response
Please see the attachment.

This manuscript is a resubmission of an earlier submission. The following is a list of the peer review reports and author responses from that submission.
Round 1
Reviewer 1 Report
The present article deals with the development and characterization of bio-based films from soy protein isolate incorporating Xanthoceras sorbifolia husk extract. Overall, the manuscript was well organized and well written and the data were sufficiently presented and discussed. However, the design of the study is simple and there are a lot of similar works describing the interaction between bio-based or protein-based films and herbal materials in packaging films. Thus, I think the data presented in this study will not add much information to our knowledge about active biobased packaging. Therefore, I can not recommend this article for publication in a high-quality journal like Foods.
Reviewer 2 Report
The paper is very well written and easy to follow. The systematic approach seems very reasonable. In my opinion, the only points that need to be addressed before publication are the following.
· The authors could provide a little more background info on SPI and XSHE (1 paragraph) for the general reader, including for example, the basic amino acid sequence present in SPI and the chemical structure of XSHE.
· I didn’t understand the superscripts in Tables 1-3. I believe they could be removed without prejudice. The reader can look at the standard deviation values to determine whether they are statistically different or not.
· Replace “oxidization” with “oxidation” (line 45, page 6)
· In Table 2, the column with the sample names is mistakenly labeled as “Figure .”
· The units for WVP should be revised. I believe they should either be “g/msPa” or “g/m2sPa”.
· “TPC” needs to be spelled-out in the text (line 37, page 4).
· It is not clear to me where the phenolic compounds released by the SPI films come from. Initially, I thought it was from XSHE, but if that was the case, the film with 0% XSHE shouldn’t show any release in Figure 4. What amino acid or compound in SPI would classify as TPC? Perhaps, the data for 0-XSHE should be used as the baseline and subtracted from all other measurements… The authors should provide details of this discussion in the text
· Figure 4 caption: please replace “accumulative” with “cumulative”
· I couldn’t understand Figure 5. The caption days “a” is TPC, but there is no indication of that in the actual Figure 5a, besides the TPC data had already been provided in Figure 4. Additionally, Figures 5a and 5b are similarly labeled but the data in them and the captions are different. I wonder if only one of the Figures (either 5a, or 5b) should be kept in the manuscript.
· Page 8, lines 42-52: DPPH and ABTS need to be spelled out earlier in the text. The authors need to provide a discussion about the ABTS results, explaining why they are drastically lower than DPPH.
Reviewer 3 Report
In the manuscript entitiled "Development of bio-based films from soy protein isolate incorporating Xanthoceras sorbifolia husk extract: physicochemical, antioxidant and food simulant release properties", authors present interesting study on development and characterization of biobased films with antioxidant properties utilizing agroindustrail waste as the source of antioxidant additive.
Overall the manucript is well written. The study is comprehensive, fairly well designed and variety of appropriate characterization techniques have been used.
1. "The crushed husks (0.5 kg) were extracted 2 times with 95% aqueous ethanol until reflux and maintained at 80 °C for 3 h and 2 h, respectively. The extracting solution was filtered and concentrated at 50 °C. Xanthoceras sorbifolia husk extract (XSHE) was stored at 4 ° C".
Why did authors follow this method of extraction ( why specific conditions?). If it is based on literature , please provide the appropriate reference/s. Also when authors write " The extracting solution was filtered and concentrated at 50 °C", what is the measure of conentration? Was the volume initial extract decraesed 1/4th or 1/2 or what? or the concentration of a particular compound in the extract measured? How long the solution was "concentrated" at 50 °C? Is it still a solution after concentration or suspension or powder?
2. It would have been helpful if authors provide characterization of the XSHE extract. What does it contain? At least provide its attributes from the literature if the authors argument is XSHE has been well characterized in the literature for not providing their XSHE characterizations.
3. What do DPPH and ABTS stand for ? please provide full forms
4, It is hard to see difference among FIIR spectra of films with different concentration of XSHE even though authors claim there is difference and interaction between protein matrix and XSHE happening.
5 What is the stability of the XSHE in the protein matrix?
